# Bullying in Students with Attention Deficit/Hyperactivity Disorder (ADHD): Analyzing Students’ Social Status and Student–Teacher Relationship Quality

**DOI:** 10.3390/ijerph22060878

**Published:** 2025-05-31

**Authors:** Sofia Mastrokoukou, Martina Berchiatti, Laura Badenes-Ribera, Laura Galiana, Claudio Longobardi

**Affiliations:** 1Department of Political and Social Studies, University of Salerno, 84084 Fisciano, Italy; smastrokoukou@unisa.it; 2Department of Behavioral Sciences Methodology, University of Valencia, 46010 Valencia, Spain; martina.berchiatti@gmail.com (M.B.); laura.badenes@uv.es (L.B.-R.); laura.galiana@uv.es (L.G.); 3Department of Psychology, University of Turin, 10123 Turin, Italy

**Keywords:** ADHD, student–teacher relationship, peer nomination, social status, bullying

## Abstract

The present study investigated how the quality of teacher–student relationships and students’ social status among peers relate to bullying experiences in children with Attention Deficit/Hyperactivity Disorder (ADHD) and those with typical development (TD). A sample of 135 students (27 with ADHD and 108 with TD; *M* = 11.37, *SD* = 1.25) participated. Using a structural equation model, we examined whether ADHD predicted students’ relationships with teachers and peers and whether these variables, in turn, predicted bullying victimization and perpetration. The model showed a good fit. Children with ADHD reported more conflictual relationships with teachers, lower peer preference, and higher social impact compared to their typically developing peers. These relationship characteristics were differentially associated with bullying outcomes, with teacher–student conflict and peer visibility emerging as social risk factors. Although preliminary, the results suggest that the relational context—particularly teacher attention and peer dynamics—may play a critical role in shaping the social experiences of students with ADHD and could have unintended consequences within peer groups.

## 1. Introduction

### 1.1. ADHD

Attention Deficit/Hyperactivity Disorder (ADHD) is a neurodevelopmental disorder with a typical childhood onset, affecting 8.4 percent of the school-age population [1,2]. The etiology of ADHD is multifactorial and controversial, and it is associated with both genetic characteristics and environmental conditions, such as prenatal and postnatal risk factors (e.g., maternal smoking during pregnancy, prematurity, low birth weight, and extreme early adversity) [3,4]. ADHD symptoms are defined by the presence of excessive inattention and/or hyperactivity–impulsivity, which remains pervasive across different settings, such as home and school, where it can lead to disruption in the classroom, problems with academic activities and difficulties in adjustment [1]. Follow-up studies indicate that the disorder persists into adolescence and adulthood in most cases (for review, see [5,6,7,8]), with 2.5 percent of adults affected by ADHD [1]. Among long-term outcomes of ADHD, there are low self-esteem, poor social function and an increased likelihood of recurrent depression in young adulthood [9,10].

#### 1.1.1. ADHD and Bullying

A recent review reported a positive association between ADHD, depressive symptoms, and bullying involvement [11]. The literature shows a high risk for children who suffer from this disorder of being involved in bullying episodes, both as bully and victim [12,13,14,15,16,17,18,19], because of their behavioral problems [16]. Bullying involvement in children with ADHD is associated with low self-control and high scores in parental reports of behavioral problems [13,14]. In addition to typical attention and social competence difficulties, children with ADHD exposed to victimization are more likely to show psychosocial problems and are at increased risk of showing depression symptoms [11]. Moreover, both bullies and victims with ADHD experience difficulties in peer relationships and are less accepted by their peers.

#### 1.1.2. Bullying

Olweus’ understanding of bullying is perhaps the most commonly cited definition in the field, yet many have treated this term in a different way in many academic or institutional contexts. Olweus [20,21] defines bullying as “a pattern in which a student is repeatedly and systematically subjected to negative actions by one or more students”. Negative actions can take the form of physical actions (e.g., hitting), verbal actions (e.g., insults), or social/relational aggression (e.g., gossiping, derogatory gestures, or social exclusion) [21,22]. Regardless of this, international organizations have provided some additional definitions that complement Olweus’ original definition. UNESCO, for example, defines bullying as “intentional, hostile or aggressive behavior, including a power imbalance, repeated over an extended period of time”, while emphasizing that bullying affects students’ right to education and well-being around the world [23]. Recent scholarship reinforces this perspective, emphasizing that bullying is shaped by broader social, cultural, and family-level contexts and that it undermines children’s psychological development, social adjustment, and educational equity [24]. Within this broader framework, bullying can also be viewed as a behavioral issue, a human rights issue, and an educational equity issue. Moreover, bullying can also be viewed as a group process [25], revealing how the social dynamics among peers involved in bullying contribute to or mitigate bullying behavior. For example, a recent meta-analysis highlighted that parental behaviors and family context play a significant role in either increasing or protecting against bullying victimization in children and adolescents [26]. In this framework, students are often assumed to fall into one of the following positions: victim, bully (e.g., the one who bullies), reinforcer (e.g., someone who supports the bully), helper (e.g., someone who helps the bully), defender (e.g., someone who supports the victim), and outsider [25,26]. It is well documented in the literature that students who violate prevailing social norms are at increased risk of bullying peers, including students with disabilities, students who are obese or overweight, and students who identify as part of a sexual or ethnic minority [27,28,29,30]. The presence of internalizing symptoms (e.g., anxiety, depression) and externalizing behaviors (e.g., aggression, hyperactivity), as well as difficulties with interpersonal relationships, are well-documented predictors of bullying victimization and bullying dynamics [31,32].

### 1.2. Peer Relationships at School

The literature has largely focused on peer relationship among children at school [33]. A close relationship with peers has positive impacts both on physical and mental health, with advantages in academic performance, social skills, and emotional and behavioral outcomes, reducing levels of stress and anxiety and enhancing school adjustment [34,35]. In addition, students who are more accepted by classmates are more preferred by their teachers [34,35,36] and score lower levels of peer victimization [35].

Regarding students with ADHD, they show specific social skills deficits and are at risk of peer relationship difficulties [33,35,37]. ADHD symptoms of inattention, hyperactivity, or impulsivity can negatively affect children’s social functioning, displaying difficulties with interactions and relationship [1,2]. Children with ADHD have problems in making friendship, have fewer friends than their mates, tend to develop poorer quality relationships, and are less sensitive to their friends’ needs and preferences [38,39,40,41]. In addition, research indicates that children with ADHD are approximately four times more likely to be rejected by them [38]. Peer rejection in children with ADHD is related to long-term academic difficulties and poor emotional adjustment in adolescence compared to their peers [40,41].

### 1.3. The Student–Teacher Relationship

The teacher serves as an attachment figure with a fundamental role in students’ development [33,42]. Inspired by attachment theory [43,44], research on the role of the student–teacher relationship (STR) has demonstrated that a positive relationship with teachers can enhance emotional security in students [45,46]. The same as responsive parents, teachers give children a secure base to explore new learnings and a safe haven in cases of need [44,46]. According to the bioecological model [47,48,49], individuals influence each other through their behavior in a context [48]. A warm and close student–teacher relationship has a positive influence on students’ emotion regulation and peer relationships [50,51] and is associated with interest in school activities [52,53,54], and autonomous motivation to defend victims in case of bullying episodes [55]. In contrast, conflictual student–teacher relationships increase the risk of peer victimization and predict conduct problems and hyperactivity/inattention symptoms [30,56].

For students with ADHD, the presence of conduct problems appears to predict poorer teacher–student relationship quality [57]. Research have shown that the relationship between children with ADHD and their teachers is generally characterized by less emotional closeness, less co-operation, and more conflict than their peers, with a negative impact on students’ school adjustment [56,57,58]. A recent study demonstrated that reducing conflict between children with ADHD and the teacher can positively enhance their emotional engagement with school, with longer-term benefits, even though the levels of stress in teachers that have to deal with children with ADHD are higher than their colleagues [56].

However, studies focusing on the impact of relationships with teachers and social status on the inclusion of children with ADHD are scarce. Teachers need increased knowledge of ADHD, as well as information about how to best integrate students with ADHD in the classroom [59]. Given the high incidence of bullying episodes in children with ADHD [60] and the association between bullying and the student–teacher relationship [11] and peer status [30,54], it seems necessary to better investigate the role of relationships with teachers and peers in the victimization of children with ADHD. This will help parents, teachers, educators, and clinicians to make decisions and take actions to enhance inclusion of children with ADHD at school and protect them against the negative effects of bullying.

### 1.4. Aims of the Study

The present study aims to examine the associations between student–teacher relationship quality, peer social status, and involvement in bullying—both as victim and perpetrator—among children diagnosed with ADHD and their typically developing peers.

Specifically, this study investigates whether aspects of the student–teacher relationship (i.e., perceived closeness, conflict, and negative expectations) and peer status (i.e., social preference and social impact) are statistically associated with bullying outcomes and whether these variables function as potential mediators in the association between ADHD status and bullying involvement.

In line with prior empirical findings [61,62] and theoretical frameworks from developmental psychopathology [63] and interpersonal systems theory [64], the following hypotheses were formulated:

**H1.** 
*ADHD status is expected to be associated with lower social preference, higher social impact, and higher levels of student–teacher conflict.*


**H2.** 
*Lower social preference and higher social impact will be associated with increased bullying involvement (victimization and perpetration).*


**H3.** 
*Greater student–teacher conflict and negative expectations will be associated with increased bullying involvement, whereas higher closeness will be associated with reduced involvement.*


**H4.** 
*The associations between ADHD and bullying involvement will be mediated by the quality of the student–teacher relationship and peer status indicators.*


A parallel multiple mediation model was used to test these hypotheses (Figure 1), as specified in Model 4 of Hayes’ PROCESS macro. All analyses were conducted using SPSS (Version 28) and Mplus (Version 8.8) [65,66].

## 2. Materials and Methods

### 2.1. Participants

This study analyzed the data of 135 primary and secondary school students recruited from six Italian mainstream primary (40.7%) and secondary schools (59.3%). The schools were selected through convenience sampling. Both children with ADHD and children with typical development (TD) were recruited from the same school. Within the schools, 27 classes were selected; there was at least one child with ADHD per class.

The children were aged between 9 and 15 years old (*M* = 11.37; *SD* = 1.25), of whom 74.8% were male (see Table 1). The mean age for children with typical development (*n* = 108) was 11.35 (*SD* = 1.24), and for children with ADHD (*n* = 27), it was 11.48 (*SD* = 1.30). The percentage of males was 72.1% for children with typical development and 80% for children with ADHD. There were no statistically significant differences in age (*t*(121) = −0.47, *p* = 0.637, Cohen’s *d* = −0.10, 95% *CI* [−0.54, 0.34]) or in gender distribution (*χ²*(1) = 0.80, Phi coefficient = −0.08; *p* = 0.372) between children with typical development and children with ADHD.

ADHD status was based on prior clinical diagnoses provided by qualified child mental health professionals (e.g., psychologists, neuropsychiatrists), as reported by parents and/or recorded in school documentation. No additional diagnostic evaluations were conducted by the research team. Although standardized diagnostic criteria (e.g., DSM-5) were presumed in clinical practice, documentation of specific criteria was not available for verification within the scope of this study.

In addition, the data of 19 teachers were also analyzed (see Table 2). In the Italian school system, it is common for teachers—particularly subject teachers in secondary education—to be assigned to more than one class [67]. As a result, the 19 participating teachers covered the 27 classes included in the study. The teachers were a mean age of 44.77 years old (*SD* = 4.96, min = 36, max = 53) and had a mean teaching experience of 14.81 years (*SD* = 7.58, min = 4, max = 40). Of them, 89.4% were female, and 84.2% were permanently employed.

### 2.2. Measures

Socio-demographic characteristics: Both teachers and students were asked to report on the socio-demographic information: current age, gender, and school grade. Also, the teachers were asked to report the number of years of teaching, number of hours per week teaching in the class, and children family status.

Presence of Attention Deficit Hyperactivity Disorder (ADHD) in children:

In Italy, an inclusive education model is used in schools, where children with ADHD are generally taught with their peers in mainstream classes [68,69]. Access to formal educational support requires a medical certificate in the form of a written or oral statement from a neuropsychiatrist or psychologist from the public health service, in accordance with the national protocol [70,71]. Once the medical documentation is received by the school, students are entitled to educational support that includes individualized adjustments based on the received medical documentation that meet the student’s needs. Although diagnosis takes place entirely outside the boundaries of the educational establishment, schools facilitate inclusive practices. Teachers are not involved in the diagnostic process but are regularly notified—in consultation with internal support staff or school psychologists—when a staff member attempts to diagnose a student. This notification allows for the development of Individual Education Plans (IEPs, *Piani Didattici Personalizzati-*PDP, in Italian) and formalizes instructional and behavioral support [70,72]. Teachers in this study were asked to report whether or not they had a student in their classroom who had been formally diagnosed with ADHD through the public health system. The question on the questionnaire was as follows: “Does the student have ADHD?” (yes, no). Given the structured communication protocols in place, teachers are generally aware of the presence of officially diagnosed conditions among their students [72].

Adolescent Peer Relations Instrument [73]: The APRI is a self-report instrument consisting of 36 items with a Likert-type response scale (1 = *never* to 6 = *every day*) which measures three types of behaviors used to bully others (physical, verbal, and social) and three ways of being targeted (physical, verbal, and social). The reliability for the social perpetration subscale (α = 0.67) is slightly below the generally accepted threshold, which suggests that findings related to this variable should be interpreted with appropriate caution. The higher the score, the greater the frequency amounts of bullying or being bullied. The score for each subscale was generated by summing the scores for the items that made up it. For this study, the reliabilities (Cronbach’s alpha) of each of the three ways of being targeted were adequate: 0.85, 0.80, and 0.83 for verbal, physical, and social victimization, respectively. The APRI was selected for this study due to its robust theoretical foundation, multidimensional approach to peer victimization and aggression, and its continued use in contemporary research examining bullying dynamics among adolescents [74,75]. In this study, the reliability of each of the three types of behaviors used to bully others assessed as internal consistency was adequate (α = 0.86 for the verbal; α = 0.78 for the physical, and α = 0.67 for the Social). Also, the reliability of each of the three ways of being targeted was adequate (α = 0.89 for verbal; α = 0.90 for physical, and α = 0.87 for social.

Student Perception of Affective Relationship with Teacher Scale [76]: The SPARTS is a self-report instrument of 25 items with a Likert-type response scale (1 = *no, that is not true* to 5 = *yes, that is true*), designed for children aged 9 to 14 years, which measures perceptions of conflict (10 items), closeness (8 items), and negative expectations (7 items) with regard to a specific teacher. When compiling the SPARTS in our study, the students were asked to refer to their “prevalent teacher” (i.e., the teacher with whom they spent the most hours per week, which, in the Italian education system, is the Italian language or science teacher). The score for each subscale was generated by summing the scores for the items that made it up. The selection of the SPARTS instrument was based on its theoretical grounding in attachment theory and its recent validation in the Italian school context. Specifically, Longobardi et al. [77] provided empirical evidence of the SPARTS’s validity and reliability in Italian pre-adolescents, supporting its use in educational and psychological research to explore student–teacher relational dynamics.

In this study, the reliability for these subscales assessed as internal consistency was adequate (α = 0.87 for the closeness; α = 0.80 for the conflict, and α = 0.56 for the negative expectations). However, the reliability of the negative expectation subscale was below the conventional threshold (α = 0.56), and, therefore, findings involving this dimension should be interpreted with caution.

Peer nomination technique (Italian version): This peer nomination questionnaire was inspired by Moreno’s sociogram techniques [78] and Coie, Dodge, and Coppotelli’s [79] sociometric strategy for assessing peer statuses in the classroom. It consists of six questions in which children have to nominate three of their peers. The questions are the following: (i) “Who would you want as a table partner?”; (ii) “Who would you want as a schoolwork partner?”; (iii) “Who would you want as a field trip buddy?”; (iv) “Who would you NOT want as a table partner?”; (v) “Who would you NOT want as a schoolwork partner?”; and (vi) “Who would you NOT want as a field trip buddy?” For each child, the sum of the positive nominations received from all peers represented their liking (L) scores. In the same way, the sum of negative nominations received by each child represented their disliking (D) scores. The L and D scores were standardized within each class (Lz and Dz) and used to compute a social preference (SP) score (Lz − Dz) and a social impact (SI) score (Lz + Dz) for each child. Ongoing psychological and educational research continues to use this method, based on solid evidence and ecological validity in naturalistic school settings. The peer nomination method continues to be used in current studies to investigate peer processes, such as loneliness, bullying, and social adjustment in school-aged children. Geukens et al. [80], for example, examined loneliness in the school context using teacher and self-reports and peer nominations, confirming the continued importance and validity of peer nomination measures for assessing children’s standing among peers and peer relationships with other children in the school setting.

### 2.3. Procedures

This research was conducted in accordance with the ethical principles of the Italian Association of Psychology (AIP) and with the formal approval of the Institutional Review Board of the University of Turin (IRB no. 118643, date: 28 February 2018). School principals authorized their institution’s participation, and written informed consent was obtained from each participating teacher.

During the initial phase, researchers collected parental consent for students’ involvement, informing families about the study’s purpose and ensuring confidentiality and voluntary participation. In the second phase, the main classroom teacher—defined as the teacher who spends at least 18 h per week in the classroom—was interviewed for each participating group, which included at least one child with an ADHD diagnosis. Each of these teachers completed five individual surveys, one for a student with a certified ADHD diagnosis and four for students with typical development, all randomly selected from those whose parents had consented. The survey package included instruments to assess sociodemographic data, ADHD status, quality of the student–teacher relationship, behavioral strengths and difficulties, and academic performance. Teachers completed these questionnaires independently during regular school hours, which took an average of about 50 min per student.

In the final phase, participating students completed peer assessments during class time, including demographic information and a peer nomination questionnaire. Written consent was obtained from each student prior to participation. To avoid bias and ensure honest responses, students were assured anonymity and confidentiality of their responses and were specifically instructed not to share or discuss their responses with classmates. No material rewards or incentives were offered for participation.

### 2.4. Data Analysis

IBM SPSS Statistics for Windows, Version 26.0 (IBM Corp., Armonk, NY, USA) and Mplus Version 8 (Muthén & Muthén, Los Angeles, CA, USA) [81] were used to perform all statistical analyses. SPSS was used to conduct descriptive analyses, normality tests, bivariate correlations, and inferential statistics (e.g., t-tests, chi-square tests). Mplus was used to test the hypothesized structural equation mediation model based on best practices in structural equation modeling suggested in the literature [65]. The use of SPSS and Mplus for psychological and educational research is prevalent in the literature, especially in studies focused on students with ADHD, teacher–student relationships, and bullying outcomes [82,83].

The data were double-entered and checked for accuracy. Then, preliminary analyses were performed. The values of kurtosis and skewness were calculated in order to check the data normality (−3 to 3 for skewness and −10 to 10 for kurtosis). All the values for univariate skewness and kurtosis for all the variables were satisfactory [84,85]. Moreover, descriptive statistics (means and standard deviations) were computed on the socio-demographic and study variables, both in the overall sample and by group (students with ADHD or students with typical development). In addition, to analyze whether there were differences between both student groups in socio-demographic variables, independent-sample *t*-tests were performed for the continuous variables, and chi-squared tests were carried out for the categorical variables. Effect sizes were calculated using Cohen’s d for independent-sample t-tests involving continuous variables, and the phi coefficient for chi-square tests involving categorical variables, following Cohen’s conventions [86,87]. Also, Pearson’s correlation coefficients were computed to obtain an overall view of the relations among the variables in the model for both the students with ADHD and those who had typical development. Cohen [86,87,88,89] established a conventional interpretation of effect sizes, in which *r* < 0.10 is considered a small effect, *r* = 0.30 is a medium-sized effect, and *r* = 0.50 is a large effect. These guidelines were used throughout this article in interpreting the results.

The model included a sequence in which ADHD affected students’ relations with teachers and students’ status, and these variables, in turn, explained bullying victimization and perpetration (see Figure 1). Therefore, the model hypothesized an indirect effect of ADHD in students on bullying dimensions mediated by relations with teachers and students’ status.

The goodness of fit for each model was assessed with several fit indexes [84,90,91], specifically, (1) the χ^2^ statistic, which is a test of the difference between the observed covariance matrix and the one predicted by the specified model; (2) the Comparative Fit Index (CFI), which assumes a non-central chi-square distribution with cut-off criteria of 0.90 or more (ideally over 0.95; [92,93]) as indicating an adequate fit; and (3) the root mean square error of approximation (RMSEA) and its 90% confidence interval. Values higher than 0.90 for the CFI or lower than 0.08 in the RMSEA are considered a reasonable fit [84], although values of 0.95 for the CFI and of 0.06 for the RMSEA are considered to be an appropriate model fit [92,93].

## 3. Results

Descriptive statistics and independent-sample *t*-tests comparing students with ADHD and those with typical development are reported in Table 3. Statistically significant group differences were found on four variables. Students with ADHD reported significantly more conflictual teacher–student relationships (*M* = 21.04, *SD* = 6.80) than students with typical development (*M* = 17.69, *SD* = 6.80), with *t*(133) = 2.29, *p* = 0.024, Cohen’s *d* = 0.49, and 95% *CI* = [0.07, 0.92]). They also experienced higher levels of social victimization (*M* = 12.20, *SD* = 6.71) than their typically developing peers (*M* = 9.63, *SD* = 5.50), with *t*(133) = 2.08, *p* = 0.040, *d* = 0.45, and 95% *CI* = [0.02, 0.87].

Regarding peer dynamics, students with ADHD demonstrated significantly lower levels of social preference (*M* = −1.75, *SD* = 1.93) than the TD group (*M* = 0.10, *SD* = 1.55), with *t*(133) = −5.27, *p* < 0.001, *d* = −1.13, and 95% *CI* = [−1.58, −0.69], as well as significantly higher levels of social impact (*M* = 0.58, *SD* = 1.17) than their typically developing peers (*M* = 0.07, *SD* = 0.90), with *t*(133) = 2.12, *p* = 0.045, *d* = 0.53, and 95% *CI* = [0.11, 0.96]. No statistically significant group differences were found for closeness, negative expectations, verbal victimization, physical victimization, or any of the perpetration dimensions. Verbal victimization did not reach conventional levels of statistical significance (*p* = 0.090), though the effect size was moderate (Cohen’s d = 0.37). This indicates a possible trend that may warrant further investigation. However, as this result was not statistically significant, interpretations should be made with caution, and future studies with larger samples are needed to clarify this pattern. Full results are presented in Table 3.

Associations among teacher–student relationship dimensions, peer social status, and bullying behaviors are displayed in Table 4 and visually summarized in Figure 2. In students with ADHD, conflict showed moderate to strong positive associations with verbal (*r* = 0.51), physical (*r* = 0.60), and social perpetration (*r* = 0.32). Closeness was negatively correlated with verbal (*r* = −0.39) and physical (*r* = −0.50) perpetration, while negative expectations were moderately associated with verbal victimization (*r* = 0.37) and physical victimization (*r* = 0.34), though these did not reach statistical significance [82,83,84,85]. Additionally, social impact was positively and significantly related to verbal perpetration (*r* = 0.44), and social preference was moderately and negatively associated with verbal perpetration (*r* = −0.33), though not statistically significant [86,87,88,89].

Among students with typical development, conflict and negative expectations were not significantly related to bullying outcomes. However, closeness was positively and significantly associated with all three forms of bullying perpetration and victimization. Social preference showed a consistent negative relationship with all forms of bullying and victimization, while social impact was positively related to verbal victimization and all three types of bullying perpetration.

Group differences in correlation patterns between interpersonal relationships and bullying behaviors were observed and are summarized in Figure 2. In the TD group, conflict was positively but moderately associated with verbal, physical, and social perpetration (e.g., *r* = 0.36 for physical). In contrast, students with ADHD showed significantly stronger associations, particularly between conflict and physical perpetration (*r* = 0.60) and between conflict and verbal perpetration (*r* = 0.51). Notably, closeness was unrelated to physical perpetration in the TD group (*r* = –0.01) but showed a strong negative correlation among students with ADHD (*r* = −0.50), suggesting that warmer teacher–student relationships may buffer aggressive behaviors more effectively in the ADHD group. Additional differences emerged for negative expectations, which were weakly correlated with victimization outcomes in TD students but showed stronger associations in the ADHD group (e.g., negative expectation and verbal victimization: NT *r* = 0.02/ADHD *r* = 0.37). These patterns indicate that relational risk factors such as conflict and distrust may play a more prominent role in bullying involvement among students with ADHD. All correlations and significant group differences are visually represented in the matrix, where only differences that reached statistical significance (*p* < 0.05) appear in the lower triangle.

The structural equation model tested whether ADHD predicts bullying perpetration and victimization via mediating variables. The model showed a good overall fit, with χ^2^(34) = 60.93, *p* = 0.003, CFI = 0.952, SRMR = 0.038, and RMSEA = 0.077 (90% *CI* [0.044, 0.107]). The measurement model revealed strong loadings for both bullying victimization (λ = 0.865 to 0.912, *p* < 0.001) and bullying perpetration (λ = 0.700 to 0.917, *p* < 0.001). The correlation between victimization and perpetration was not statistically significant (*r* = 0.270, *p* = 0.076).

ADHD was a significant negative predictor of social preference (β = −0.416, *p* < 0.001) and a positive predictor of social impact (β = 0.210, *p* = 0.025) and conflict (β = 0.195, *p* = 0.019). Victimization was directly predicted by lower social preference (β = −0.152, *p* = 0.028), higher conflict (β = 0.421, *p* < 0.001), and higher closeness (β = 0.168, *p* = 0.021). Indirect effects of ADHD on victimization were also significant (β = 0.164, 95% *CI* [0.075, 0.274], *p* = 0.001). Perpetration was directly predicted by higher social impact (β = 0.205, *p* < 0.001) and higher conflict (β = 0.445, *p* < 0.001), with significant indirect effects from ADHD (β = 0.182, 95% *CI* [0.070, 0.317], *p* = 0.002). These results indicate that students with ADHD, who tend to experience more conflict with teachers and higher social visibility, are more likely to engage in bullying behavior. Correlations among the mediating variables are presented in Table 5.

## 4. Discussion

The present study examined how students with ADHD perceive their relationships with teachers and classmates and how these perceptions relate to experiences of bullying, both as victims and perpetrators. Although the intersection of ADHD, social dynamics, and school adjustment has been examined previously [9,10,11,14], this study contributes a new perspective by examining bullying within a broader relational framework, highlighting not only the behavior itself but also the structural and emotional contexts in which it occurs.

One of the most striking findings of this study is that students with ADHD reported significantly more conflict in their relationships with teachers than their typically developing peers. This is consistent with a body of research linking ADHD symptoms to greater relationship strain in the classroom [30,35,36,42,56,57]. However, our findings also suggest that such conflict may have additional social effects beyond the teacher–student dyad. Conflictual teacher relationships emerged as a pronounced risk factor for bullying perpetration among students with ADHD. This aligns with prior research and underscores how adversarial interactions with authority figures may spill over into peer dynamics. This pattern highlights a possible relational pathway through which negative interactions with authority figures may spill over into peer relationships and shape the way students with ADHD engage in and experience the social world of the classroom.

In addition to teacher-related conflict, indicators of peer relationships also showed that notable disparities were observed. Students with ADHD faced markedly lower peer acceptance compared to their typically developing classmates. This social marginalization, combined with their heightened visibility in peer groups, may amplify vulnerability to bullying. This combination—low likability but high visibility—deserves careful consideration. While high social influence may indicate that students with ADHD are conspicuous among their peers, it does not necessarily mean that they are accepted. The finding that social preference was lower supports the view that visibility does not equate to inclusion. Instead, it may reflect a form of social conspicuousness that puts these students at greater risk of being scrutinized and excluded by their peers. This could partly explain the increased levels of social victimization reported by this group.

Although the associations between certain dimensions of the teacher relationship and bullying outcomes did not always reach statistical significance, several correlations showed moderate effect sizes [86,87,88,89], suggesting potentially significant trends. For example, among students with ADHD, negative expectations of the teacher relationship were moderately related to verbal and physical victimization. Although these results did not reach statistical significance and should be interpreted with caution, they may reflect a trend. Given the low internal consistency of the negative expectations subscale, these associations should be interpreted with particular caution.

The comparison with students with TD offers additional possibilities for interpretation. In this group, teacher conflict and negative expectations were not significantly associated with bullying outcomes. Contrary to expectations, teacher closeness was not significantly related to either bullying perpetration or victimization, suggesting a more limited role of closeness in bullying dynamics among typically developing students in this sample (Table 4). This finding is somewhat counterintuitive, given that teacher closeness is generally seen as a protective factor. One possible explanation is that visible teacher–student closeness may be perceived by classmates as favoritism, potentially eliciting peer resentment or envy. Such dynamics may contribute to social exclusion or tension, particularly in adolescent peer cultures where perceived fairness is highly valued. These interpretations are consistent with prior work on peer responses to perceived teacher partiality [51,55]. Future studies should investigate whether it is the perception—and not merely the presence—of closeness that contributes to these peer outcomes. This finding may reflect the complexity of peer dynamics in adolescence, in which proximity to a teacher can be perceived as either a source of support or a social burden, depending on the context. In addition, indicators of peer status were particularly meaningful for typically developing students: students with higher social preference were consistently less likely to engage in bullying, while higher social influence was associated with more aggressive behaviors. These patterns highlight the nuanced ways in which peer status can act as both a protective and a risk factor.

The structural model highlighted the role of ADHD as a predictor of bullying through relational mediators. ADHD status was associated with greater teacher conflict, higher social impact, and lower social preference—factors that, in turn, were related to bullying outcomes. While these associations align with the proposed model, the cross-sectional nature of the data precludes conclusions about causality.

Interestingly, the model also indicated that teacher–student closeness was associated with higher levels of bullying victimization. While this may seem counterintuitive—given that closeness is typically viewed as a protective factor—it is possible that students who are perceived as particularly close to their teachers may become more visible or stand out in the classroom. This could lead to peer resentment or perceptions of favoritism, especially during adolescence when peer approval is highly valued. However, this interpretation is speculative, as peer perceptions of favoritism were not directly assessed in the current study.

Taken together, the results suggest a relational ecology in which students with ADHD are particularly vulnerable to social difficulties not only because of their behavioral symptoms but also because of the ways in which these symptoms interact with teacher expectations, peer perceptions, and group dynamics. Elevated peer visibility among students with ADHD, especially when reinforced by teacher attention—however well-intentioned—may inadvertently contribute to their social marginalization. This aligns with frameworks such as social comparison theory and peer group status theory [94,95], which emphasize how perceived inequalities in attention or affiliation with peers can foster resentment or exclusion.

This study also encourages reflection on the ambivalent role of teacher proximity. Although often seen as uniformly beneficial, the findings suggest that proximity to teachers in certain peer contexts may have unintended social costs. This does not diminish the value of close teacher–student relationships but rather requires a calibrated approach that takes into account the broader ecosystem of the classroom and the potential for different peer interpretations of teacher behavior.

In summary, this study adds to a growing body of evidence that ADHD is not only a cognitive or behavioral disorder but also a profoundly social one. The way students with ADHD are perceived and treated—both by teachers and peers—plays a critical role in shaping their school experiences. By taking a more holistic view of classroom relationships, researchers and practitioners alike can better understand and address the complex mechanisms underlying bullying in this population.

### 4.1. Limitations and Future Research

Some limitations of the present work should be discussed. The main limitation of this study is the small subsample of children with ADHD, which may have reduced the likelihood of detecting statistically significant associations (i.e., statistical power) and consequently limits the precision and generalizability of the findings. Moreover, as aforementioned, given this small sample size, it was not possible to analyze the potential moderating role of the ADHD variable in the relationships between the student–teacher relationship, students’ peer status, and bullying dimensions.

Another limitation relates to the characteristics of the sample (e.g., the mono-cultural setting), which may constrain the generalizability of the findings. In this regard, cross-cultural studies comparing different cultural groups and school settings using similar measures and variables could contribute to a more comprehensive understanding of these associations. Furthermore, the sample was drawn from a limited geographic context, which may not reflect broader variability in school climate, teacher practices, or peer norms. As a result, these findings may not be representative of children and teachers in other contexts, such as urban areas or culturally diverse populations.

In addition, this study employed a convenience sampling method, which involves non-random selection of participants. This approach may limit the representativeness of the sample and introduce selection bias. Future research would benefit from employing more systematic sampling procedures to enhance external validity.

Moreover, ADHD diagnoses were based on external clinical assessments reported by parents or schools. This study did not verify the use of standardized diagnostic criteria (e.g., DSM-5), which introduces uncertainty regarding the consistency of ADHD classification across participants. This limitation should be taken into account when interpreting group-level comparisons.

The data analyzed in this study were based on self-report measures, which depend on participants’ perceptions and willingness to respond accurately. Social desirability bias may have affected some responses and contributed to measurement variability. Including an appropriate measure of social desirability in future studies could allow for its statistical control in the analysis (e.g., as a covariate).

Another limitation concerns the internal consistency of two subscales: the social perpetration dimension of the APRI (α = 0.67) and the negative expectations dimension of the SPARTS (α = 0.56). These values are below the commonly accepted threshold for adequate reliability. As such, findings related to these subscales should be interpreted with caution. Future studies could consider revising or expanding the item sets to improve measurement quality.

Finally, the cross-sectional design of this study does not allow for inferences about the directionality of associations among variables. Although mediation analyses were conducted, the application of such models to cross-sectional data is subject to interpretative limitations [50]. Longitudinal studies are needed to examine how these relationships develop over time. In addition, future research would benefit from multi-informant designs (e.g., peer nominations, teacher ratings) to enhance the validity of self-reported data.

### 4.2. Practical and Policy Implications

This study advances the field by highlighting a previously under-researched consequence of teacher support in classes with students with ADHD: the possibility that visibility and relational closeness inadvertently contribute to social risks within the peer group. This nuanced insight suggests that interventions should go beyond the standard goal of increased teacher support and consider the complex peer dynamics that may result.

While teacher closeness is often seen as a protective factor, our findings suggest scenarios where teacher attention may provoke envy or social distancing among peers, particularly during adolescence when peer status becomes increasingly important. This adds a relational layer to intervention planning, especially for educators working in inclusive classrooms.

This has practical implications for the training of teachers, who need to be aware of the subtle ways in which support can influence group dynamics. A balance between inclusive teaching and equal peer engagement is essential. In addition, the findings highlight the importance of whole-class interventions aimed at building emotional literacy, reducing stigma, and promoting group cohesion.

Programs that promote peer empathy, collaborative learning, and positive behavioral norms can help reduce the social costs of “otherness” or individual attention. From a policy perspective, our findings suggest that individualized support strategies (e.g., PDPs) should be embedded within a broader relational framework that considers not only the student–teacher dyad but also the peer ecosystem. Teacher training modules could explicitly address how well-intentioned attention might be perceived within the peer group and how to mitigate unintended consequences.

Interventions that promote an inclusive peer culture may be more effective in addressing bullying than those that focus solely on behavior correction or the teacher–student relationship. Finally, this work contributes to a growing body of evidence urging schools to view ADHD not only through a clinical or cognitive lens, but as a disorder with profound social implications that requires systemic responses at educational, psychological, and relational levels. In doing so, educators and policymakers can foster environments where social dynamics are proactively managed, and inclusivity becomes a shared classroom value.

## 5. Conclusions

This study set out to explore how students—particularly those with ADHD—perceive their relationships with teachers and peers, and how these perceptions relate to their experiences of bullying, both as victims and perpetrators. The results provide preliminary yet meaningful evidence that ADHD-related behaviors are embedded within broader relational dynamics in the classroom, influencing social standing and bullying involvement.

Using this method, it was found that teacher–student conflict and peer visibility may be risk factors for students with ADHD. Students self-reported that they experienced conflictual relationships with teachers, increased levels of social victimization, and increased social visibility and impact on peer dynamics. In relation to the structural model of the study, the data show indirect pathways of ADHD behavioral patterns in relation to classroom victimization and perpetration, suggesting that social effects of neurodevelopmental traits occur in more nuanced interpersonal spaces.

Limitations of cohort size, self-reported data, and cross-sectional data prevent causal statements and generalizations. Nevertheless, the findings suggest a different picture of relational coordination rather than a unidimensional view, as teachers’ attention is often well-meaning to support a student, but they may inadvertently exacerbate an already weak peer behavior or social structure, leading to further feelings of isolation or alienation.

Overall, this research provides a point of reference if we are interested in considering the social ecology of the classroom in supporting students with ADHD. Future studies that incorporate longitudinal and peer-reported data, as well as data from culturally diverse families, will deepen the understanding of neurodevelopmental differences in the relational spaces of school. Until then, the findings provide evidence base for educators and school psychologists developing social inclusion and educational success for children with ADHD status.

## Figures and Tables

**Figure 1 ijerph-22-00878-f001:**
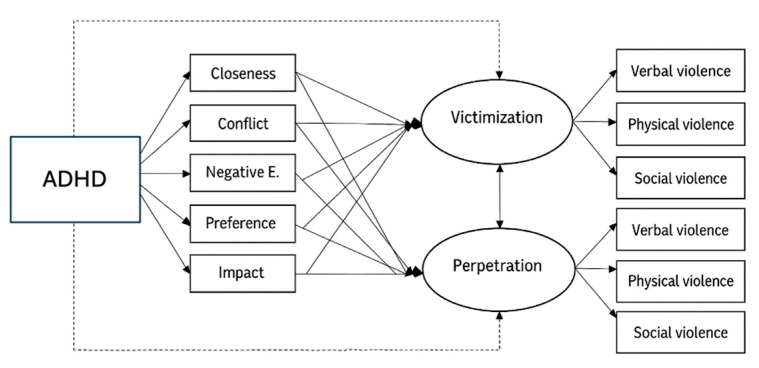
Structural equation model predicting bullying victimization and perpetration. Notes: Correlations between bullying dimensions and among closeness, conflict, negative expectations, social preference (preference), and social impact (impact) were estimated. Discontinued arrows show the direct effects of ADHD on the bullying dimensions of victimization and perpetration.

**Figure 2 ijerph-22-00878-f002:**
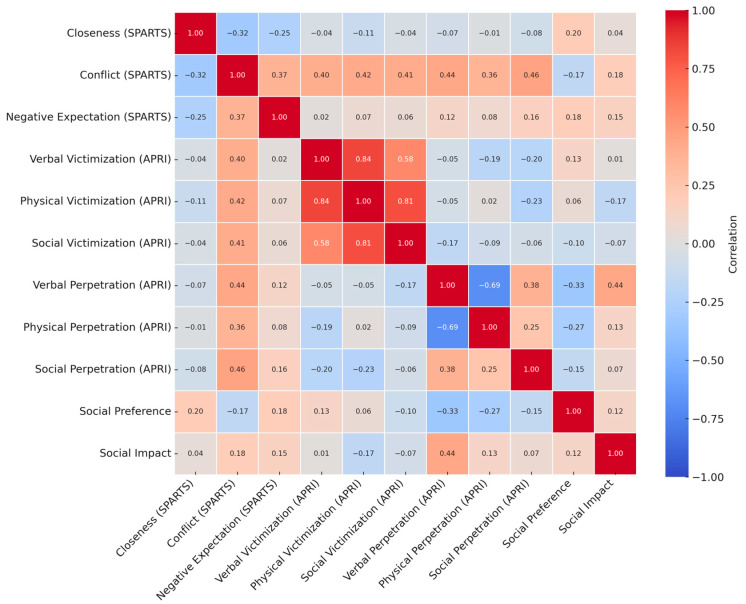
*Correlation matrix for students with typical development (TD, upper triangle) and significantly differing correlations in students with ADHD (lower triangle).* ***Note*.** The upper triangle displays Pearson’s r coefficients for the TD group. The lower triangle shows only those correlations where the ADHD group significantly differs from the TD group (*p* < 0.05), based on Fisher’s r-to-z transformation. Values are presented in the format: TD r/ADHD r. Empty cells in the lower triangle indicate no significant group difference. Color intensity reflects the strength of correlation in the TD group. The red dashed line on the scale indicates the interpretive threshold (*r* = ±0.38) for *n* = 27.

**Table 1 ijerph-22-00878-t001:** Descriptive information for students with ADHD by class.

Class ID	Grade	Gender	Age	Socioeconomic Status	Years Since ADHD Diagnosis
C01	Primary 4	Male	9	Medium	1
C02	Primary 4	Female	9	Low	1
C03	Primary 5	Male	10	Medium	2
C04	Primary 5	Male	10	High	2
C05	Primary 6	Male	11	Low	2
C06	Primary 6	Male	11	Medium	3
C07	Secondary 1	Male	12	High	3
C08	Secondary 1	Female	12	Medium	3
C09	Secondary 2	Male	13	Low	4
C10	Secondary 2	Male	13	Medium	4
C11	Secondary 3	Male	14	Medium	5
C12	Secondary 3	Female	14	High	5
C13	Secondary 3	Male	14	Low	5
C14	Primary 4	Male	9	Low	1
C15	Primary 5	Female	10	Medium	2
C16	Primary 6	Male	11	High	2
C17	Secondary 1	Male	12	Medium	3
C18	Secondary 1	Male	12	Low	3
C19	Secondary 2	Female	13	Medium	4
C20	Secondary 2	Male	13	High	4
C21	Secondary 3	Male	14	Medium	5
C22	Secondary 3	Female	14	Medium	5
C23	Primary 5	Male	10	Low	2
C24	Primary 6	Female	11	High	2
C25	Secondary 1	Male	12	Medium	3
C26	Secondary 2	Male	13	Low	4
C27	Secondary 3	Male	15	High	5

**Table 2 ijerph-22-00878-t002:** Teacher–class assignments and demographics.

Teacher ID	Class IDs	Gender	Age	Years of Teaching	Socioeconomic Info	Type of Employment
T01	C01, C02	Female	45	15	Medium	Permanent
T02	C03	Female	42	14	Medium	Permanent
T03	C04	Male	48	20	High	Permanent
T04	C05	Female	40	10	Low	Permanent
T05	C06	Female	44	13	Medium	Temporary
T06	C07	Female	47	16	High	Permanent
T07	C08	Female	50	22	Medium	Permanent
T08	C09	Female	46	17	Low	Permanent
T09	C10	Male	36	4	Medium	Temporary
T10	C11	Female	43	14	Medium	Permanent
T11	C12	Female	49	18	High	Permanent
T12	C13	Female	53	25	Low	Permanent
T13	C14	Female	38	8	Low	Temporary
T14	C15	Female	44	11	Medium	Permanent
T15	C16	Male	41	10	High	Permanent
T16	C17	Female	46	13	Medium	Permanent
T17	C18	Female	39	6	Low	Temporary
T18	C19, C20	Female	48	19	Medium	Permanent
T19	C21, C22, C23, C24, C25, C26, C27	Female	51	30	Mixed	Permanent

**Table 3 ijerph-22-00878-t003:** Mean (*SD*) scores for whole sample and students’ groups (students with Attention Deficit Hyperactivity Disorder [ADHD] and students with typical development [TD]) on all variables and *t* test.

		Total Sample	Students with ADHD	Students with TD			
	*Range*	*M (SD)*	*M (SD)*	*M (SD)*	*t*	*p*	*Cohen’s d (95% CI)*
Closeness (SPARTS)	8–40	28.13 (7.91)	29.78 (8.68)	27.71 (7.69)	1.22	0.225	0.26 (−0.16, 0.69)
Conflict (SPARTS)	10–44	18.36 (6.91)	21.04 (6.80)	17.69 (6.80)	2.29	0.024	0.49 (0.07, 0.92)
Negative Expectations (SPARTS)	7–28	15.36 (4.92)	15.94 (4.67)	15.21 (4.99)	0.69	0.490	0.15 (−0.27, 0.57)
Verbal Victimization (APRI)	6–36	11.72 (6.57)	13.64 (7.21)	11.24 (6.34)	1.71	0.090	0.37 (−0.06, 0.79)
Physical Victimization (APRI)	6–32	9.41 (5.59)	10.57 (6.66)	9.11 (5.29)	1.21	0.228	0.26 (−0.16, 0.68)
Social Victimization (APRI)	6–36	10.15 (5.83)	12.20 (6.71)	9.63 (5.50)	2.08	0.040	0.45 (0.02, 0.87)
Verbal Perpetration (APRI)	6–29	9.30 (3.82)	9.78 (3.95)	9.18 (3.80)	0.73	0.466	0.16 (−0.26, 0.58)
Physical Perpetration (APRI)	6–25	8.24 (3.49)	8.61 (3.98)	8.15 (3.68)	0.62	0.534	0.13 (−0.29, 0.56)
Social Perpetration (APRI)	6–28	10.25 (3.76)	9.96 (4.05)	10.32 (3.70)	−0.44	0.664	−0.09 (−0.52, 0.33)
Social Preference (Z scores)	--	−0.27 (1.79)	−1.75 (1.93)	0.10 (1.55)	−5.27	<0.001	−1.13 (−1.58, −0.69)
Social Impact (Z scores)	--	0.17 (0.97)	0.58 (1.17)	0.07 (0.90)	2.12	0.045	0.53 (0.11, 0.96)

Note. *CI* = confidence interval for effect size. SPARTS = Student Perception of Affective Relationship with Teacher Scale. APRI = Adolescent Peer Relations Instrument.

**Table 4 ijerph-22-00878-t004:** Inter-correlations between all variables with results for students with Attention Deficit Hyperactivity Disorder in the top diagonal (upper triangle) and for students with typical development in the bottom diagonal (lower triangle).

	1	2	3	4	5	6	7	8	9	10	11
1. Closeness (SPARTS)	–	−0.32	−0.05	0.25	0.19	0.14	−0.39 *	−0.50 **	−0.12	0.22	0.01
2. Conflict (SPARTS)	−0.32 **	–	0.25	0.14	0.18	0.21	0.51 ***	0.60 ***	0.32	−0.06	0.26
3. Negative Expectation (SPARTS)	−0.25 **	0.37 ***	–	0.37	0.34	0.19	0.10	0.10	0.29	0.03	0.15
4. Verbal Victimization (APRI)	−0.04	0.40 ***	0.02	–	0.84 ***	0.58 **	−0.05	−0.19	−0.20	0.13	0.01
5. Physical Victimization (APRI)	−0.11	0.42 ***	0.07	0.82 ***	–	0.68 ***	−0.05	0.02	−0.23	0.06	−0.17
6. Social Victimization (APRI)	−0.04	0.41 ***	0.06	0.86 ***	0.81 ***	–	−0.17	−0.09	−0.06	−0.10	−0.07
7. Verbal Perpetration (APRI)	−0.07	0.44 ***	0.12	0.42 ***	0.50 ***	0.43 ***	–	0.69 ***	0.38 *	−0.33	0.44 *
8. Physical Perpetration (APRI)	−0.01	0.36 ***	0.08	0.43 ***	0.62 ***	0.40 ***	0.73 ***	–	0.25	−0.27	0.13
9. Social Perpetration (APRI)	−0.08	0.46 ***	0.16	0.44 ***	0.48 ***	0.49 ***	0.73 ***	0.62 ***	–	0.15	0.07
10. Social Preference	0.20 *	−0.17	0.18	−0.23 *	−0.20 *	−0.23 *	−0.24 *	−0.18	−0.19 *	–	−0.54 **
11. Social Impact	0.04	0.18	0.15	0.20 *	0.10	0.16	0.26 **	0.30 **	0.22 *	0.12	–

Note. * *p* < 0.05. ** *p* < 0.01. *** *p* < 0.001. SPARTS = Student Perception of Affective Relationship with Teacher Scale. SDQ = Strengths and Difficulties Questionnaire. APRI = Adolescent Peer Relations Instrument.

**Table 5 ijerph-22-00878-t005:** Correlations among closeness, conflict, negative expectations, preference, and impact in the structural equation model.

	1	2	3	4	5
1. Closeness (SPARTS)	–				
2. Conflict (SPARTS)	−0.320 **	–			
3. Negative Expectations (SPARTS)	−0.209 **	0.346 ***	–		
4. Social Preference	0.205 *	−0.145	0.144	–	
5. Social Impact	0.032	0.200 **	0.149	−0.068	–

*Note*. * *p* < 0.05; ** *p* < 0.01; *** *p* < 0.001.

## Data Availability

The data presented in this study are available on request from the corresponding author due to privacy reasons.

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
