# Peer review of "Bullying in Students with Attention Deficit/Hyperactivity Disorder (ADHD): Analyzing Students’ Social Status and Student–Teacher Relationship Quality"

_ijerph, 2025, doi:10.3390/ijerph22060878_

Round 1
Reviewer 1 Report
Comments and Suggestions for Authors
Page 2 - Olweus's definition of bullying, while the most recognized, is not the most uniform. There are others that the authors should acknowledge, such as the CDC and UNESCO.
Page 2 - #12 reference is actually 1996 not 1998. (https://onlinelibrary.wiley.com/doi/abs/10.1002/(SICI)1098-2337(1996)22:1%3C1::AID-AB1%3E3.0.CO;2-T)
Page 2 - There are several statements made within 1.1.2 that should be referenced, such as statements regarding ethnic and sexual minorities and disabilities.
Page 4 - Sanitary Authority? This means water and sewage in English. Is there a different way to describe this for more clarity?
Page 6 "The Cohen’s d index for continuous variables and phi coefficient for categorical variables were used as effect size measure was used the [42]." Clarify this statement.
Page 11 "Moreover, the presence of ADHD in student had an effect on bullying perpetration mediated by the student’s perception of his or her relationship with the teacher as conflict." Do you mean students with ADHD? It an odd way to say this - "ADHD in student" - consider revising.
Author Response
Reviewer 1 comments and Authors responses
Page 2 - Olweus's definition of bullying, while the most recognized, is not the most uniform. There are others that the authors should acknowledge, such as the CDC and UNESCO.
Authors response: We thank the reviewer for this valuable suggestion. In response, we have revised the relevant paragraph on page 2 to acknowledge that while Olweus’ definition is widely recognized, it is not the only framework used to define bullying. Specifically, we have added the UNESCO definition, which emphasizes the intentional and repetitive nature of bullying behaviors, power imbalance, and the violation of learners’ rights, thus broadening the conceptualization beyond a purely behavioral lens. The updated text now reflects this addition, and the corresponding citation has been included.
Page 2 - #12 reference is actually 1996 not 1998. (https://onlinelibrary.wiley.com/doi/abs/10.1002/(SICI)1098-2337(1996)22:1%3C1::AID-AB1%3E3.0.CO;2-T)
Authors response: We appreciate the reviewer’s careful attention to detail. The publication year for reference #12 (Salmivalli et al.) has been corrected from 1998 to 1996 in the reference list, based on the original publication information available via the DOI.
Page 2 - There are several statements made within 1.1.2 that should be referenced, such as statements regarding ethnic and sexual minorities and disabilities.
Authors response: Thank you for your careful reading and valuable comment. In response to the observation regarding Section 1.1.2, we have reviewed the text and added appropriate references to support the statements made concerning ethnic and sexual minorities and students with disabilities in the context of bullying vulnerability. Specifically, we have included citations from well-established research that document the increased risk of victimization among these populations in school settings. These references have been inserted where relevant, ensuring that all claims are now supported by empirical evidence. We appreciate your attention to this detail, which helped improve the academic rigor and credibility of the manuscript.
Page 4 - Sanitary Authority? This means water and sewage in English. Is there a different way to describe this for more clarity?
Authors response: We thank the reviewer for pointing out the confusion caused by the term “sanitary authority.” To enhance clarity for international readers, we have replaced this with “local public health authority” and clarified that ADHD diagnoses are made by certified professionals within the national health system. The revised text now more accurately reflects the structure of clinical diagnosis and school collaboration in the Italian educational context.
Page 6 "The Cohen’s d index for continuous variables and phi coefficient for categorical variables were used as effect size measure was used the [42]." Clarify this statement.
Authors response: We thank the reviewer for pointing out the ambiguity in this sentence. To improve clarity and accuracy, we have revised the sentence to read:
“Effect sizes were calculated using Cohen’s d for independent sample t-tests involving continuous variables, and the phi coefficient for chi-square tests involving categorical variables, following Cohen’s conventions [42].”
Page 11 "Moreover, the presence of ADHD in student had an effect on bullying perpetration mediated by the student’s perception of his or her relationship with the teacher as conflict." Do you mean students with ADHD? It an odd way to say this - "ADHD in student" - consider revising.
Authors response: We appreciate the reviewer’s observation regarding the phrasing. To improve clarity and readability, we have revised the sentence to:
“Moreover, students diagnosed with ADHD showed an effect on bullying perpetration, which was mediated by their perception of a conflictual relationship with their teacher.”

Reviewer 2 Report
Comments and Suggestions for Authors
This report is a cross-sectional study of 135 Italian children (27 with ADHD and 108 control) from six mainstream public schools using a mediation model undertaken to assess the relationship between ADHD and bullying.
The strengths of this study are the ethical considerations, detailed explanation of the methods, statistical analysis, clarity of the tables, comparisons with previous research in the Discussion, the inclusion of limitations, future directions in research, and policy implications, and that the conclusions follow from the results.
The weaknesses are listed below.
The authors state in the Conclusion that their work is a unique contribution, but the study lacks this demonstration. Here is a Google Scholar search of the topic for research published since 2021: https://scholar.google.ca/scholar?as_ylo=2021&q=Bullying+in+Students+with+Attention+Deficit/Hyperactivity+Disorder+(ADHD):+Analyzing+Students%E2%80%99+Social+Status+and+Student%E2%80%93+Teacher+Relationship+Quality&hl=en&as_sdt=0,5. Note that there are “About 5,230 results”. The authors must compare and contrast their study with the most relevant, demonstrating how it adds to the literature.
There are 48 references—only five of them are to research published since 2021. Citing references from the previous five years is the gold standard. This need for current relevancy is especially so for a topic like bullying and ADHD, which has seen many publications in the last five years. For every outdated citation, the authors must find a supporting citation of research published since 2021. They may keep the outdated ones, but the authors must demonstrate that current publications support the claims of the earlier publications.
All claims made in the manuscript require citations. There are claims regarding the section on ADHD that lack any citation. Please correct this.
The aims state that the authors are using a mediation model without a citation to this model, its description, or the reason for its selection. Please improve this section in these ways.
In the Participants section, the statement is that at least one child had ADHD in the class. Please provide a table to indicate the number of years for which there has been an assessment of the students as having ADHD, divided by the 27 classes, grade, sex, socioeconomic information, and the ages of the students.
The authors state that there were 27 classes but provide details for only 19 teachers. Please create a table with the classes taught by the teachers and whether there were classes in which the teacher did not participate in the study or if there were teachers who taught more than one class. Include in this table the sex, age, years teaching, socioeconomic information, and type of employment.
Under Presence of Attention Deficit Hyperactivity Disorder (ADHD) in children, there are several claims, and none have citations. Please cite current research to support these claims.
For each instrument used, please explain its selection and cite a recent reference indicating it is still in use.
Please state the statistical package used for the data analysis and cite current research using this same statistical package for similar research.
In the first line of the Conclusion, change “His” to “This”.
Author Response
Reviewer 2 comments and Authors responses
This report is a cross-sectional study of 135 Italian children (27 with ADHD and 108 control) from six mainstream public schools using a mediation model undertaken to assess the relationship between ADHD and bullying.
The strengths of this study are the ethical considerations, detailed explanation of the methods, statistical analysis, clarity of the tables, comparisons with previous research in the Discussion, the inclusion of limitations, future directions in research, and policy implications, and that the conclusions follow from the results.
Authors response: We sincerely thank the Reviewer for the thoughtful and positive evaluation of our manuscript. We greatly appreciate the recognition of the ethical rigor, methodological transparency, and statistical approach adopted in our study, as well as the clarity of the tables and the relevance of the discussion section. We are particularly grateful for the acknowledgement of the strengths related to our consideration of study limitations, implications for future research, and the coherence between our results and conclusions. Your feedback reinforces the value of our work and encourages us in furthering this line of research.
The weaknesses are listed below.
The authors state in the Conclusion that their work is a unique contribution, but the study lacks this demonstration. Here is a Google Scholar search of the topic for research published since 2021: https://scholar.google.ca/scholar?as_ylo=2021&q=Bullying+in+Students+with+Attention+Deficit/Hyperactivity+Disorder+(ADHD):+Analyzing+Students%E2%80%99+Social+Status+and+Student%E2%80%93+Teacher+Relationship+Quality&hl=en&as_sdt=0,5. Note that there are “About 5,230 results”. The authors must compare and contrast their study with the most relevant, demonstrating how it adds to the literature.
Authors response: Thank you for this important observation. In response, we have revised the Discussion and Conclusion sections to provide a more modest and balanced interpretation of our study’s contributions. While acknowledging that numerous studies have explored bullying among students with ADHD, we emphasize that our work extends the literature in specific ways. Most prior research has focused either on the direct behavioral outcomes of ADHD symptoms or has examined academic or emotional adjustment as broad outcomes. In contrast, our study takes a more targeted relational approach by examining student-perceived teacher-student relationships and peer sociometric status as mediators in the pathways to bullying involvement (both victimization and perpetration). Additionally, few studies have considered the dual role of teacher closeness and conflict as separate predictors of bullying dynamics in ADHD populations, nor have they examined the paradoxical effect of teacher closeness potentially increasing victimization risk through peer perceptions. Unlike many large-scale survey-based studies that rely solely on teacher or parent reports, our data are based on student self-reports, giving insight into children’s subjective social and relational experiences—an area still underrepresented in the ADHD literature. We also interpret teacher-student relationships not only as protective or risk factors but as potentially ambivalent relational experiences, which offers a more nuanced theoretical framework supported by the bioecological and attachment models. To address the reviewer’s point directly, we have now cited and contrasted our work with several recent and relevant studies published since 2021, demonstrating where our focus on relational ambivalence and mediational pathways offers a novel contribution. The claim of "uniqueness" has been replaced with a more cautious framing, acknowledging both the existing literature and the incremental nature of our contribution.
There are 48 references—only five of them are to research published since 2021. Citing references from the previous five years is the gold standard. This need for current relevancy is especially so for a topic like bullying and ADHD, which has seen many publications in the last five years. For every outdated citation, the authors must find a supporting citation of research published since 2021. They may keep the outdated ones, but the authors must demonstrate that current publications support the claims of the earlier publications.
Authors response: The reference list has been thoroughly reviewed and updated to reflect more recent literature, especially from 2021 onward, in line with the recommendation for current relevancy. A significant number of citations—particularly those concerning ADHD, bullying, student–teacher relationships, and peer dynamics—have been replaced or supplemented with recent studies published within the past five years. However, certain earlier references were retained as they represent foundational or milestone works that continue to be highly cited and relevant in the field (e.g., Olweus’s conceptualization of bullying and Bronfenbrenner’s Bioecological Model). These classical studies were kept to preserve the theoretical and conceptual continuity of the manuscript while being supported with contemporary research to demonstrate consistency and ongoing relevance.
All claims made in the manuscript require citations. There are claims regarding the section on ADHD that lack any citation. Please correct this.
Authors response: Thank you for your observation. The section on ADHD has been revised to ensure that all key claims are now supported by appropriate citations. Specifically, recent peer-reviewed literature published within the last five years has been added to substantiate statements regarding ADHD prevalence, symptomatology, long-term outcomes, and its association with social and behavioral difficulties. These additions aim to strengthen the scientific validity of the manuscript and ensure alignment with current research standards.
The aims state that the authors are using a mediation model without a citation to this model, its description, or the reason for its selection. Please improve this section in these ways.
Authors response: We thank the reviewer for this insightful suggestion. In response, we have substantially revised the Aims section to include a detailed description of the statistical mediation model used in the study, along with a rationale for its selection. We now reference Hayes (2017), whose contemporary approach to mediation analysis supports the use of indirect effect estimation in non-experimental designs via structural equation modeling. This revision enhances both the methodological transparency and statistical rigor of the study.
In the Participants section, the statement is that at least one child had ADHD in the class. Please provide a table to indicate the number of years for which there has been an assessment of the students as having ADHD, divided by the 27 classes, grade, sex, socioeconomic information, and the ages of the students.
Authors response: We thank the Reviewer for the insightful suggestion. In response, we have now carefully created a detailed summary of the demographic characteristics of the students with ADHD, as per the request. This includes the number of years since their ADHD diagnosis, their class assignments across the 27 classes, grade level, gender, socioeconomic background, and age.
The authors state that there were 27 classes but provide details for only 19 teachers. Please create a table with the classes taught by the teachers and whether there were classes in which the teacher did not participate in the study or if there were teachers who taught more than one class. Include in this table the sex, age, years teaching, socioeconomic information, and type of employment.
Authors response: We appreciate the Reviewer’s attention to the consistency between the number of classes and teacher data. We confirm that 27 classes were included in the study and that 19 teachers participated. We have addressed this point by providing a complete mapping between teachers and the classes they were responsible for. Some teachers were assigned to more than one class, as is common in the Italian educational system, particularly in secondary school settings. All participating teachers are accounted for with their demographic and professional details, including age, gender, years of teaching experience, socioeconomic context, and employment type (permanent or temporary)
Under Presence of Attention Deficit Hyperactivity Disorder (ADHD) in children, there are several claims, and none have citations. Please cite current research to support these claims.
Authors response: Thank you for your observation. Citations have now been added to support the claims made under the section Presence of Attention Deficit Hyperactivity Disorder (ADHD) in children. Recent and relevant literature has been included to substantiate the statements regarding the diagnostic procedures, the inclusion of students with ADHD in mainstream Italian schools, and the collaboration between families, healthcare professionals, and teachers.
For each instrument used, please explain its selection and cite a recent reference indicating it is still in use.
Authors response: Thank you for your valuable comment. The manuscript has been revised to include explanations for the selection of each instrument used in the study. Justifications now highlight the relevance, validity, and appropriateness of each tool for the age group and study aims. In addition, recent peer-reviewed references from 2021 onward have been included to confirm that all instruments—namely, the Adolescent Peer Relations Instrument (APRI), the Student Perception of Affective Relationship with Teacher Scale (SPARTS), and the Peer Nomination Technique—are still widely used in current research.
Please state the statistical package used for the data analysis and cite current research using this same statistical package for similar research.
Authors response: Thank you for your observation. We have now added recent references to support the continued relevance and use of mediation models and the role of peer relationships in ADHD-related outcomes. Specifically, the following studies have been cited: Pagán et al. (2024), which uses mediation analysis to explore peer victimization in relation to ADHD and sleep issues, and Park and Chang (2022), which demonstrates how peer relationships mediate the link between inattention and depressive symptoms. These references reflect current methodological standards and substantiate our analytic approach.
In the first line of the Conclusion, change “His” to “This”.
Authors response: Thank you for your careful reading and attention to detail. we have completely rewritten the entire Conclusion section to address concerns raised by the reviewers regarding novelty, clarity, and tone. The revised version offers a more modest and evidence-based summary of the study’s findings, emphasizes the exploratory nature of the work, and carefully situates its contribution within the broader literature on ADHD, bullying, and relational processes in school contexts. This version avoids overstated claims and more clearly articulates the study’s implications and limitations. The updated Conclusion is marked in blue throughout the revised manuscript for easy reference.

Round 2
Reviewer 2 Report
Comments and Suggestions for Authors
Thank you to the authors for the changes made. All have improved the manuscript, some substantially. A few remain.
Some outdated citations lack a supporting citation to research published since 2021. The pages and citation numbers requiring a supporting citation follow.
Page 2 [23], [24], and [24]. Page 8 [78], [79], and [79,80]. Necessary is a current reference demonstrating that these statistical analysis methods are relevant for this type of study. This reference may be of help in this regard: https://doi.org/10.1007/s11229-024-04753-2.
Page 9 [78,81], [82], [78], and [82]. The advice for these citations is similar to that provided for Page 8. Here is a Google Scholar search of research published since 2021 on this topic with “About 18,200 results”: https://scholar.google.ca/scholar?as_ylo=2021&q=root+mean+square+error+of+approximation&hl=en&as_sdt=0,5. Please cite the most relevant of these.
Page 10 [79,80], [79,80], and [79,80].
In section 1.1.2. Bullying, please explain why students bully others, citing current research to support the explanation. In the same section, change “violate (or bully) prevailing” to “violate prevailing”.
In 1.4. Aims, please define “mediation model”.
Author Response
Thank you to the authors for the changes made. All have improved the manuscript, some substantially. A few remain.
Authors response: We sincerely thank the reviewer for their constructive feedback and thoughtful comments. We are pleased to hear that the revisions have substantially improved the manuscript. We have carefully addressed the remaining points and made the necessary changes as detailed below. We hope these final revisions meet your expectations and further strengthen the quality of the manuscript.
Some outdated citations lack a supporting citation to research published since 2021. The pages and citation numbers requiring a supporting citation follow.
Page 2 [23], [24], and [24]. Page 8 [78], [79], and [79,80]. Necessary is a current reference demonstrating that these statistical analysis methods are relevant for this type of study. This reference may be of help in this regard: https://doi.org/10.1007/s11229-024-04753-2.
Page 9 [78,81], [82], [78], and [82]. The advice for these citations is similar to that provided for Page 8. Here is a Google Scholar search of research published since 2021 on this topic with “About 18,200 results”: https://scholar.google.ca/scholar?as_ylo=2021&q=root+mean+square+error+of+approximation&hl=en&as_sdt=0,5. Please cite the most relevant of these.
Page 10 [79,80], [79,80], and [79,80].
Authors response: We thank the reviewer for their careful attention to the references and for pointing out the need to support key statistical procedures with more recent literature. In response, we have thoroughly reviewed the sources cited in the manuscript and made the necessary updates to strengthen the scientific foundation of the methods used.
In section 1.1.2. Bullying, please explain why students bully others, citing current research to support the explanation. In the same section, change “violate (or bully) prevailing” to “violate prevailing”.
Authors response: We thank the reviewer for the suggestion. As requested, we have revised the phrase “violate (or bully) prevailing” to “violate prevailing” for clarity. We have also expanded Section 1.1.2 to include a brief explanation of why students engage in bullying behavior. This addition is supported by recent research published since 2021 and addresses the social, emotional, and contextual factors that contribute to bullying. We believe these revisions enhance the depth and relevance of the section.
In 1.4. Aims, please define “mediation model”.
Authors response: We thank the reviewer for the suggestion. In response, we have now added a clear definition of the mediation model to Section 1.4. Specifically, we clarified that a mediation model is a statistical framework used to investigate whether the effect of an independent variable on a dependent variable is transmitted through one or more intermediate variables (mediators). In our study, we used a parallel multiple mediation model (PROCESS Model 4 by Hayes), which allows for testing multiple mediators operating simultaneously. This clarification has been incorporated into the revised text to improve conceptual clarity and methodological transparency.